# Effect of 10-Day Treatment with 50 mg Prednisolone Once-Daily on Haemostasis in Healthy Men—A Randomised Placebo-Controlled Trial

**DOI:** 10.3390/biomedicines11072052

**Published:** 2023-07-21

**Authors:** Peter Kamstrup, Ema Rastoder, Pernille Høgh Hellmann, Pradeesh Sivapalan, Emil List Larsen, Jørgen Vestbo, Charlotte Suppli Ulrik, Jens P. Goetze, Filip Krag Knop, Jens Ulrik Stæhr Jensen

**Affiliations:** 1Section of Respiratory Medicine, Department of Medicine, Copenhagen University Hospital—Herlev and Gentofte, 2900 Hellerup, Denmark; peter.kamstrup@regionh.dk (P.K.);; 2Center for Clinical Metabolic Research, Department of Medicine, Gentofte Hospital, University of Copenhagen, 2900 Hellerup, Denmark; 3Department of Clinical Biochemistry, Copenhagen University Hospital—Rigshospitalet, 2100 Copenhagen, Denmark; 4Allergi og Lungeklinikken Vanløse, 2720 Vanløse, Denmark; 5Division of Infection, Immunity and Respiratory Medicine, School of Biological Sciences, The University of Manchester, Manchester Academic Health Science Centre, Manchester M13 9PL, UK; 6Department of Respiratory Medicine, Copenhagen University Hospital—Hvidovre, 2650 Hvidovre, Denmark; 7Department of Clinical Medicine, Faculty of Health and Medical Sciences, University of Copenhagen, 2200 Copenhagen, Denmark; 8Department of Biomedical Sciences, Faculty of Health and Medical Sciences, University of Copenhagen, 2200 Copenhagen, Denmark; 9Steno Diabetes Center Copenhagen, 2730 Herlev, Denmark

**Keywords:** randomised controlled trial, thromboelastography, glucocorticoids, healthy volunteers, haemostasis, biomarkers

## Abstract

Synthetic corticosteroids are widely used due to their anti-inflammatory and immunosuppressant effects. Their use has been associated with venous thromboembolism, but it is unknown whether thromboembolism has a causal relationship with corticosteroid treatment. In a randomised, double-blind, placebo-controlled trial in normal to overweight healthy men, the effect of the corticosteroid prednisolone on haemostasis using either 50 mg prednisolone or matching placebo once daily for ten days was investigated. The primary outcome was a change from baseline in the viscoelastic measurement maximal amplitude of clot in kaolin-activated thromboelastography (TEG). Changes from baseline in other TEG measurements, D-dimer, von Willebrand factor (VWF) antigen, and ristocetin cofactor activity (RCo), antithrombin, protein C, prothrombin, fibrinogen, INR, APTT, and platelet count were secondary outcomes. Thirty-four men participated in this study. Compared to placebo, prednisolone treatment did not affect maximal amplitude of clot (difference −0.77 (95% confidence interval (CI) −2.48, 0.94) mm, *p =* 0.37, missing: *n* = 2), but it altered VWF antigen (28%, *p =* 0.0004), VWF:RCo (19%, *p =* 0.0006), prothrombin (5%, *p =* 0.05), protein C (31%, *p <* 0.0001), antithrombin (5%, *p =* 0.013), and fibrinogen (−15%, *p =* 0.004). Thus, prednisolone treatment did not alter TEG-assessed maximal amplitude of clot, despite that it affected prothrombotic markers (increased prothrombin, VWF antigen, VWF:RCo, prothrombin, and decreased fibrinogen) and increased antithrombotic markers (protein C and antithrombin).

## 1. Introduction

Synthetic corticosteroids are commonly used as anti-inflammatory drugs to treat various conditions, including, but not limited to, chronic obstructive pulmonary disease, asthma, inflammatory bowel disease, and rheumatoid diseases [1]. Reported side effects of corticosteroids include hyperglycaemia, infections, osteoporosis, and a variety of cardiovascular disorders [2,3,4,5]. Observational studies have shown an association between the use of corticosteroids and venous thromboembolism, including pulmonary embolism [5,6] as well as cardiovascular and cerebrovascular events [7,8]. Whether this association is due to underlying diseases or corticosteroid treatment is uncertain.

Randomised controlled trials investigating the effect of corticosteroids on haemostasis without underlying inflammation are sparse. The main focus has been on primary haemostasis, where increased VWF levels were reported in two of four studies [9,10,11,12]. In one study, platelet levels were investigated and increased by 17% after high-dose dexamethasone treatment [11]. Secondary haemostasis parameters are more inconsistently investigated: The markers of the intrinsic and extrinsic pathways, APTT and INR, respectively, have not been reported. However, Brotman et al. reported increased levels of factors VIII (+27%) and XI (+6%), both parts of APTT, as well as increased levels of factor VII (+13%), which is a part of INR [10]. They also found an increase in fibrinogen because of corticosteroid treatment [10]. Additionally, thrombin has been found to be increased following prednisolone [9]. Conversely, D-dimer, the product of fibrin degeneration, does not appear to change due to corticosteroid treatment [10,12].

Viscoelastic measurements provide summarised comprehensive information on whole-blood samples, and in addition, different assays allow for investigation of each pathway [13]. Interestingly, one study showed that endogenous hypercortisolism (i.e., Cushing’s syndrome), compared to matched controls, was associated with shorter intrinsic clotting time as assessed using viscoelastic testing (159.3 s vs. 172.2 s), while other intrinsic and extrinsic viscoelastic measurements were unaffected [14]. Furthermore, studies on conventional haemostasis parameters have been carried out, including in patients with endogenous hypercortisolism, with inconsistent results on most parameters, except for VWF:Ag, which was found to be increased [15,16,17,18,19,20,21]. However, these studies are also subject to bias arising from the selection of patients and the underlying disease. Observational studies have been conducted in patients with underlying inflammation, such as patients with systemic lupus erythematosus and polymyalgia rheumatica [22,23,24,25]. A meta-analysis found that use of corticosteroids in underlying inflammation reduces VWF:Ag, VWF:RCo, and fibrinogen while increasing antithrombin, protein C, and protein S [23]. These studies may nevertheless be unsuited for describing the isolated effect of corticosteroids, as their impact on inflammation may alter coagulation [26] and the corticosteroids may alter the inflammation [27].

Here, using a randomised placebo-controlled design, we investigated how the most used synthetic corticosteroid, prednisolone, affects viscoelastic measurements of coagulation in healthy men. We also evaluated the effect of prednisolone on a variety of static coagulation parameters across primary and secondary haemostasis as well as regulatory mechanisms.

## 2. Materials and Methods

### 2.1. Trial Design

The current study is a sub-study of the CURPRED (“The effect of curcumin on the development of prednisolone-induced hepatic insulin resistance in overweight and obese participants”) trial was a randomised, controlled, double-blind trial investigating the effect of curcumin on prednisolone-induced glucometabolic side effects [28]. After an interim analysis, evaluation of the curcumin intervention was halted for futility, since it was judged that curcumin did not affect the glucometabolic side effects, and the trial was completed using two arms (prednisolone/placebo). Using these two arms, we wanted to elucidate the unclear effect of prednisolone on the haemostasis of healthy individuals, using kaolin-TEG measurements alongside conventional haemostatic parameters.

### 2.2. Participants

Participants were recruited through posters, advertisements in newspapers, and a national webpage (https://www.forsoegsperson.dk/, accessed on 30 September 2022) for persons wanting to participate in research projects. The primary study included male participants with a written informed consent, body mass index (BMI) > 24.9 kg/m^2^, haemoglobin ≥ 7.5 mmol/L, aged between 18 and 59 years, and excluded participants on antidiabetic treatment, lipid-lowering drugs, clopidogrel, oral anticoagulant treatment, frequent use of nonsteroid anti-inflammatory drugs in the last two months, use of medication known to interact with prednisolone, use of curcumin-containing food supplements, use of any regular medication that could not be discontinued for 18 h, intake of more than 21 units of alcohol per week, known liver or kidney disease, known type 2 diabetes, or HbaA1c above 48 mmol/mol at screening, participation in- or planned lifestyle changes, contraindication for magnetic resonance imaging, or any other condition that, per investigator judgment, could interfere with trial participation or safety of participants. When the curcumin intervention was halted, eligibility criteria were broadened to facilitate recruitment: The BMI criterion was lowered to >20.0 kg/m^2^, the HbA1c cut-off was lowered to 42 mmol/mol, and the exclusion criteria of magnetic resonance imaging contra indication and use of curcumin-containing food supplies were removed. Furthermore, persons with medically treated asthma, hypertension, current smokers, and persons not vaccinated for COVID-19 were excluded.

### 2.3. Sample Size

As outlined in our statistical analysis plan, estimation was based on TEG-assessed maximal amplitude using a paired *t* test. The assumptions for the power calculation were a level of significance of 5%, a power of 80%, two-sided statistics, a detectable difference of 4.0 mm, and a standard deviation of 4.0 mm. With a 1:1 distribution between active treatment and placebo, this resulted in a sample size of at least 32. Using the observed data, a post-hoc power calculation was computed with the abovementioned level of significance and power.

### 2.4. Randomisation and Treatment

The first 14 participants were randomised (1:1:1) to either (a) prednisolone 50 mg once daily and curcumin 100 mg twice daily for 10 days, (b) prednisolone 50 mg once daily and curcumin placebo twice daily for 10 days, and (c) prednisolone placebo once daily and curcumin placebo twice daily for 10 days. Treatments were blinded to participants and investigators recruiting and allocating participants. To balance the treatment groups, the last 20 participants were randomised in a 1:3 ratio to active prednisolone treatment and 2:3 to placebo. Each treatment was numbered individually, ensuring blinding of patients as well as investigators. Unblinding of investigators conducting the present sub-study was carried out following study completion of the last participant. The randomisation sequences were made by scientists not affiliated with this study, using a random sequence generator. The Pharmacy of the Capital Region of Denmark supplied the prednisolone and prednisolone placebo as encapsulated tablets to ensure double blinding.

### 2.5. Procedures and Analyses

This study was conducted at Gentofte Hospital, University of Copenhagen. Following a 10 h overnight fasting period, each participant had blood samples taken prior to randomisation and at end of treatment (24 h following the last dose). Each of the visits were planned at the same time of day. Sampling was carried out by cubital vein cannulation. As TEG samples are known to be especially sensitive to preanalytical factors, researchers performing blood collections were trained to conduct this uniformly. TEG samples were citrated whole-blood samples (3.2% sodium citrate) and were analysed within 2 h of withdrawal. All other analyses except for platelets were performed on sodium citrate plasma (3.2%). Platelets were measured in K_2_-EDTA plasma.

Seventeen pairs of the fibrinogen and antithrombin samples were subject to one extra freeze–thaw cycle. All pairs of activated partial thromboplastin clotting time (APTT), 12 pairs of prothrombin, 4 pairs of D-dimer, and 2 pairs of international normalised ratio (INR) were run on frozen 3.2% sodium citrate plasma (−80 °C for samples stored >3 months, −20 °C for samples stored <3 months). Prior to freezing, biobank samples were centrifuged for 15 min at 2900× *g* and 4 °C. Frozen samples were thawed using standardised procedures. TEG samples were analysed using the Kaolin TEG assay (TEG^®^ 5000 Hemostasis analyzer, Haemonetics, Boston, MA, USA). All samples were analysed within 120 min of collection. VWF:Ag was measured using the HemosIL^®^ von Willebrand Factor Antigen assay. von Willebrand factor ristocetin cofactor activity (VWF:RCo) was measured using the HemosIL^®^ von Willebrand Factor Ristocetin Cofactor Activity assay. Protein C was measured using the HemosIL^®^ Protein C assay. Prothrombin was measured using the HemosIL^®^Readiplastin, HemosIL^®^Factor II deficient plasma assay using an ACL TOP 550 (Instrumentation of Laboratory, Werfen Company, Barcelona, Spain). Antithrombin was measured using the Siemens INNOVANCE Antithrombin assay. D-dimer was measured using the Siemens INNOVANCE D-dimer assay. APTT was measured using the Siemens Dade Actin FS assay. Fibrinogen was measured using the Siemens Dade thrombin assay. INR was measured using the MediRox Owrens PT assay using a Siemens Sysmex CS5100 (Siemens Healthcare Diagnostics, Marburg, Germany). Platelet count was measured using Siemens ADVIA 120 CBC TIMEPAC and ADVIA 120 SHEAT/RINSE on a Siemens ADVIA 2120i 3M (Siemens Healthcare Diagnostics, Marburg, Germany).

### 2.6. Outcomes

The primary outcome was a change from baseline in maximal amplitude of clot (measured using TEG, TEG:MA). Secondary outcomes were changes from baseline in TEG-assessed reaction time (R) (TEG:R), clot lysis in 30 min (TEG:LY30), clot strengthening (angle) (TEG:Angle), and clot formation (K) (TEG:K) as well as platelet count, and levels of VWF:Ag, VWF:RCo, INR, APTT, prothrombin, fibrinogen, D-dimer, protein C, and antithrombin. All outcomes were collected and assessed after the final participant completed the study.

### 2.7. Statistical Analyses

Each of the biomarkers was examined using a constrained linear mixed model including time as a fixed variable and assuming an unstructured covariance pattern to account for replicate measurements on the same subject. If baseline data showed a skewness in either BMI (difference > 3 kg/m^2^) or age (difference > 10 years), adjustment was performed for the variable in question as both additive and multiplicative variables. As no skewness was observed, no further analyses were conducted. To minimise influence by possible effects of curcumin, we conducted a planned ancillary analysis to evaluate any effect of the curcumin intervention. As any participant failing to comply would be replaced with another participant, all analyses were performed as intention-to-treat. A two-sided *p* value of 0.05 was considered statistically significant. Spaghetti plots depicting the paired measurements per marker of each participant were shown for each of the endpoints. A statistical analysis plan was published online prior to unblinding and analysis of results (http://coptrin.dk/wp-content/uploads/2023/01/sap-signeret.pdf (accessed on 14 January 2023)). Data management, descriptive statistics, and constrained linear mixed modelling were performed using Statistical Analysis Software 9.4 (SAS Institute, Cary, NC, USA). Illustration of graphics were created using R 4.1.2 (R Foundation for Statistical Computing, Vienna, Austria) with the survival 3.2–13, ggplot2 3.4.2, and patchwork 1.1.2 packages.

## 3. Results

We screened 39 potential participants from May 2020 to December 2022, of whom 34 completed the study with 17 randomised to prednisolone and 17 to placebo treatment (Figure 1 and Table 1). There were no missing data on baseline information. All participants were compliant to medicine intake according to the predefined criteria (i.e., >70%). A total of 15 participants experienced adverse events (8 in the prednisolone group and 7 in the placebo group). Although no serious adverse events were registered, one participant in the combined active prednisolone and placebo curcumin group suffered from epistaxis and terminated the study treatment on day 8 as per investigator instruction. The trial was completed when 34 participants finished the study, as per our sample size estimate.

### 3.1. Primary Outcome

Compared with placebo, TEG:MA was not affected by prednisolone treatment: −0.77 ((95% confidence interval (CI) −2.48, 0.94) mm, *p =* 0.37) (Table 2 and Figure 2).

### 3.2. Secondary Outcomes

We found that prednisolone, when compared to placebo, increased VWF:Ag by 28% (0.31 (95% CI 0.15, 0.46) kIU/L, *p =* 0.0004), VWF:RCo by 19% (0.15 kIU/L (95% CI 0.07, 0.23 kIU/L), *p =* 0.0006), prothrombin by 5% (0.05 kIU/L (95% CI −0.001, 0.11 kIU/L), *p =* 0.05), protein C by 31% (0.30 kIU/L (95% CI 0.19, 0.41 kIU/L), *p <* 0.0001), and antithrombin by 5% (0.05 kIU/L (95% CI 0.01, 0.09 kIU/L), *p =* 0.013). Fibrinogen decreased by 15% (−1.02 µmol/L (95% CI −1.69, −0.36 µmol/L), *p =* 0.004) in the prednisolone-treated group compared to in the placebo-treated group. Neither TEG:R, TEG:LY30, TEG:Angle, nor TEG:K changed following prednisolone treatment compared to placebo. Likewise, APTT, INR, D-dimer, and platelet count were not affected by prednisolone treatment. Outcomes are illustrated in Figure 2, Figure 3, Figure 4 and Figure 5, and Table 2 provides a detailed description of the regression analyses alongside information on missing data. 

### 3.3. Ancillary Curcumin Analysis

Curcumin possibly modified the effect of prednisolone on platelet count, see Appendix A. Neither primary nor other secondary outcomes were affected by curcumin treatment.

### 3.4. Post-Hoc Power Calculation

A post-hoc power calculation using a level of significance of 0.05, sample size of 15 in the prednisolone group, 17 in the placebo group, and the obtained standard deviation (SD) 3.0 mm revealed that due to the lower-than-expected SD we were able to, with a power of 0.80, detect a difference in TEG:MA of 3.1 mm.

## 4. Discussion

In this randomised double-blind placebo-controlled trial, we found that 10-day prednisolone treatment (50 mg QD) in healthy men did not affect global viscoelastic measurement of intrinsic coagulation but affected markers of both primary and secondary haemostasis, with increased levels of VWF antigen, VWF activity, prothrombin, protein C, and antithrombin, as well as decreased levels of fibrinogen.

Prednisolone treatment increased the concentration of markers in the primary haemostasis, as both VWF antigen and VWF activity increased. For VWF antigen, evidence from previous RCTs is conflicting, which is highlighted in a meta-analysis [10,11,12,23]. We found increased levels of VWF antigen, a conclusion that is supported by the more recent Majoor study [9]. VWF activity has, to our knowledge, not been investigated in RCTs, and the evidence from observational studies on endogenous hypercortisolism is conflicting [15,21]. Overall, our findings suggest increased activity of primary haemostasis in response to prednisolone treatment.

We observed no changes in markers of the extrinsic nor intrinsic pathway (both INR and APTT were unchanged in the prednisolone-treated group), consistent with a previous RCT [10]. In the common pathway, we found a slight increase in prothrombin in the prednisolone group, a marker that had not been investigated in this context prior to the current study. Additionally, we found that fibrinogen decreases in prednisolone treatment, which contradicts prior knowledge [10]. Despite the changes in prothrombin and fibrinogen, D-dimer levels did not seem to change in the prednisolone-treated group, as previously observed [9,10,12].

Protein C and antithrombin, two regulators of the coagulation pathways that had not been investigated in RCTs previously, both increased in response to prednisolone treatment.

Given our finding that TEG measurements were not affected by prednisolone is not a type 2 error, this could indicate that increased levels of the antithrombotic factors may compensate for the increased levels of prothrombotic factors in healthy individuals. This is supported by the knowledge that important genetic risk factors for VTE are antithrombin deficiency and the *prothrombin 20210A*-mutation [29]. Lack of antithrombin increases the risk of VTE [30]. Likewise, the *prothrombin 20210A*-mutation increases risk of VTE [31] through even slight increases in the levels of prothrombin, albeit without functionally damaging the protein [29]. Nevertheless, our data do not support testing for thrombophilia during or immediately following corticosteroid treatment.

When we measure blood markers, the measurement reflects the combination of biosynthesis, release, and elimination/consumption. Thus, our findings are most likely a combination of changes in these factors. An illustration of this is knowledge on VWF, where investigators in a previous in vitro study observed that VWF biosynthesis, measured by cellular mRNA expression, was increased when high-dose dexamethasone was applied; however, dexamethasone alone was not sufficient to stimulate release from the cells [11]. Nevertheless, dexamethasone increased the thrombin-mediated release of VWF from vascular endothelial cells [32,33]. We found a decrease in fibrinogen as a response to prednisolone, and this might reflect either a decrease in biosynthesis, release, or a combination. Alternatively, it might result from an increased consumption of fibrinogen. In support of the latter, it has been reported that glucocorticoids may increase expression of fibrinogen genes [34]. Our antithrombin results are in line with previous in vivo findings showing that antithrombin expression increases in patients receiving dexamethasone [35].

The main result of our trial, which was that we found no changes in the maximum clot strength in kaolin-TEG, could suggest no overall change in the strength of clots following activation of the intrinsic cascade. It does, however, not rule out a change in other TEG-parameters or in the extrinsic pathway. Changes did occur in conventional haemostatic measurements with an apparent increase in primary haemostasis activity as well as changes in both the common pathway and regulatory mechanisms of secondary haemostasis, which suggests a complex interplay of coagulation when affected by prednisolone. Even though we found no effect on clot strength in the intrinsic pathway, it is not possible for us to rule out any other thrombogenic effect of prednisolone, especially via the extrinsic cascade. However, it is important to note that corticosteroids are commonly used for treatment of underlying inflammatory disorders, which themselves may lead to increased thrombogenicity [36,37]. In this situation, corticosteroids may reduce thrombogenicity by reducing systemic inflammation. Lastly, in daily clinical practise, concomitant treatment with corticosteroids may also be part of alterations in the haemostatic system.

Future studies should include evaluation of the extrinsic cascade and further aim to implement real time measurements (e.g., bleeding time) before, during, and after treatment. Further, it is important to note that the structure and function of the vessel walls also influence haemostasis and thrombogenicity in the body [38], which is not included in our study.

A strength of the current study is its randomised double-blind placebo-controlled design. As far as we know, this is the first trial investigating the effect of synthetic corticosteroids on viscoelastic measurements. Furthermore, we investigated several important haemostasis parameters of both primary and secondary haemostasis including a dynamic measurement of the intrinsic pathway, and the effect of prednisolone on several of these markers has not previously been described in healthy individuals. All participants were fasting at both sampling times, and to reduce the influence of the circadian rhythm, participants were seen at the same time of day. Our study also has limitations. Firstly, although the sample size was not large, we did adhere to the predefined power calculation. Furthermore, the post-hoc power calculation revealed that even a smaller difference than expected could be identified, due to the lower SD. However, since our power measurement was performed on TEG:MA, the other outcomes may be subject to coincidental findings and should be evaluated accordingly, as the sample size of the study is limited in its size, this includes the other TEG-measurements, as TEG:MA has a lower variability than the other measurements [39]. Secondly, we had missing data from two patients on the primary outcome parameter; these were lost during transportation of the samples, and thus were missing completely at random. Thirdly, eligibility criteria were changed during the study, to increase the recruitment potential. The influence of this change did not introduce bias, as measurements were before and after measurements on the same participants, and as expected, according to the randomisation, the baseline measurements were balanced between the groups. Fourthly, the fixed dose for a fixed duration prevented us from investigating a dose–response relationship. Fifthly, five participants were subject to a concomitant curcumin intervention. Since curcumin did not affect the primary outcome, it is not likely that our results were influenced by this as also indicated by ancillary analysis. Lastly, as previously described, and only for secondary outcomes, samples for a few (17 fibrinogen and antithrombin) measurements underwent an extra freeze–thaw cycle, and samples for some measurements (all pairs of APTT, 12 pairs of prothrombin, 4 pairs of D-dimer, and 2 pairs of INR) were analysed after storage in a biobank (this was nevertheless only carried out on markers previously evaluated for stability during long term freezing [40] and analyses were always carried out on both samples of the participant and followed standardised procedure).

In conclusion, functional coagulation of the intrinsic pathway, as measured using TEG:MA, did not appear to be affected by corticosteroid therapy. Our analysis of the conventional prothrombotic and antithrombotic measures may explain the seemingly neutral effect of corticosteroids on functional coagulation; however, these were secondary outcomes, and this study was not necessarily powered to conclude on the basis of these measurements.

Our results do not encourage changes in prescription patterns.

## Figures and Tables

**Figure 1 biomedicines-11-02052-f001:**
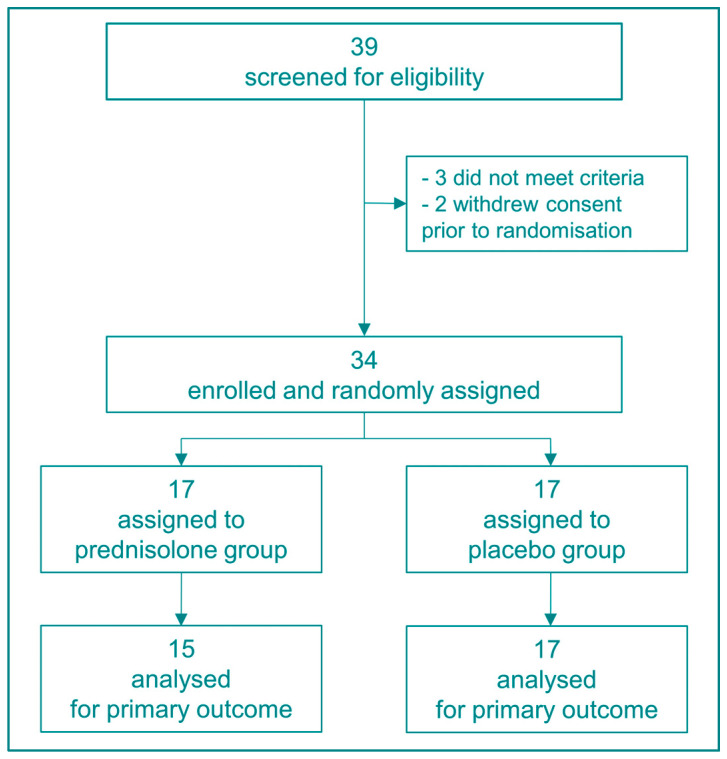
Study flowchart.

**Figure 2 biomedicines-11-02052-f002:**
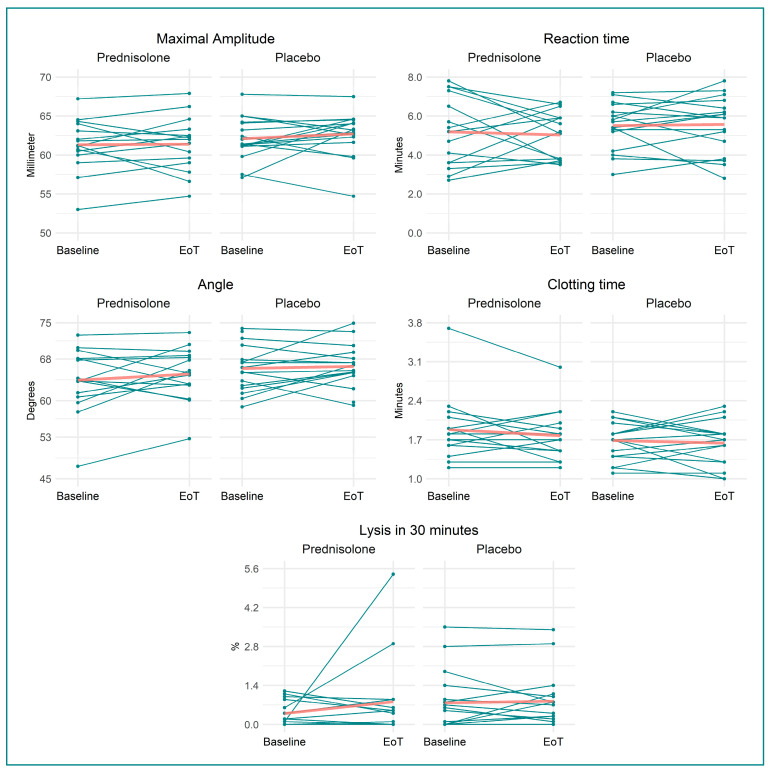
Spaghetti and scatter plots for the thromboelastography measurements split into the two treatment groups—before and after treatment. Red line denotes mean at baseline and EoT. Abbreviations: EoT: end of treatment.

**Figure 3 biomedicines-11-02052-f003:**
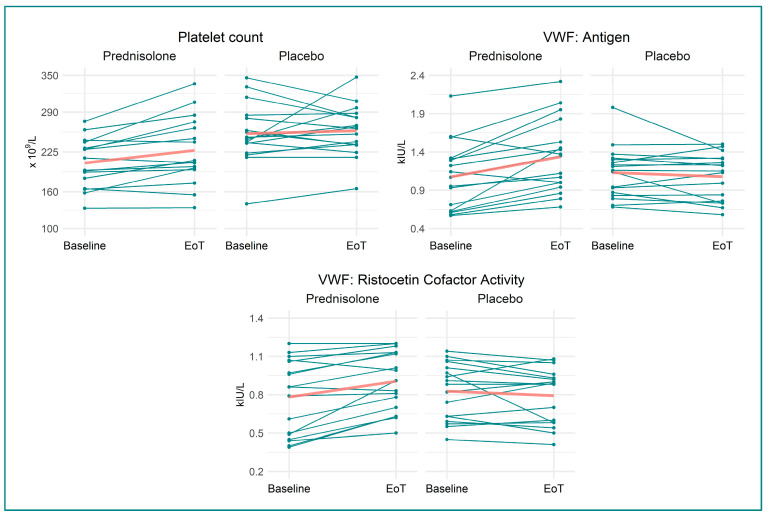
Spaghetti and scatter plots for the primary haemostasis measurements. Split into the two treatment groups—before and after treatment. Red line denotes mean at baseline and EoT. Abbreviations: EoT: end of treatment; VWF: von Willebrand factor; kIU/L: kilo international units per litre.

**Figure 4 biomedicines-11-02052-f004:**
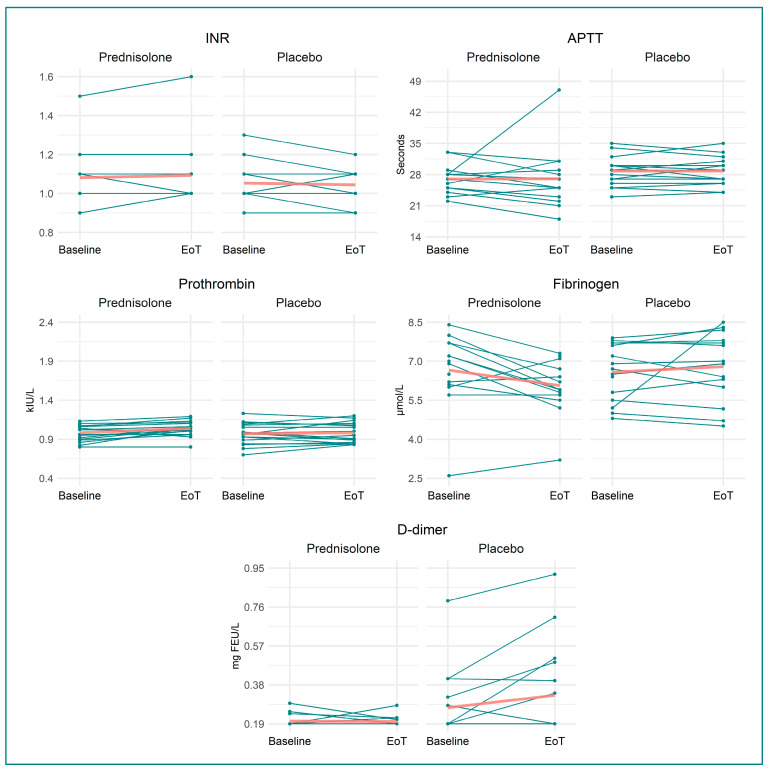
Spaghetti and scatter plots for the secondary haemostasis measurements. Split into the two treatment groups—at baseline and end of treatment. Red line denotes mean at baseline and EoT Abbreviations: INR: international normalised ratio; EoT: end of treatment; APTT: activated partial thromboplastin clotting time; kIU/L: kilo international units per litre; FEU: fibrin equivalent unit.

**Figure 5 biomedicines-11-02052-f005:**
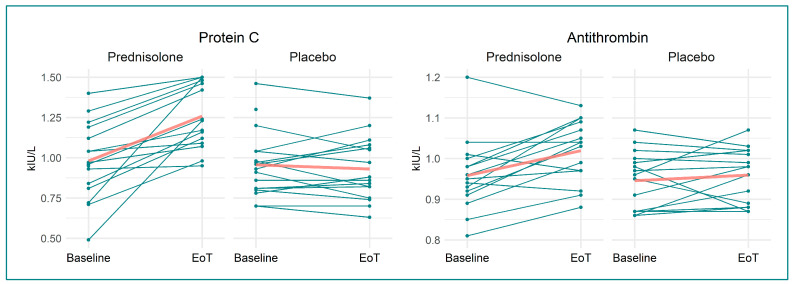
Combined spaghetti and scatterplots for the regulatory mechanism measurements. Split into the two treatment groups—at baseline and end of treatment. Red line denotes mean at baseline and EoT. Abbreviations: kIU/L: kilo international units per litre; EoT: end of treatment.

**Table 1 biomedicines-11-02052-t001:** Baseline characteristics.

	Placebo (*n* = 17)	Prednisolone (*n* = 17)
Sex, male, *n* (%)	17 (100.0)	17 (100.0)
Age, years, mean (SD)	29.2 (8.3)	33.6 (11.5)
BMI, kg/m^2^, mean (SD)	25.7 (2.4)	26.6 (2.7)
Ethnicity		
Caucasian, *n* (%)	17 (100.0)	14 (82.4)
Asian, *n* (%)	0 (0.0)	3 (17.7)
Former smoker, *n* (%)	1 (5.9)	3 (17.7)
Diabetes, *n* (%)	1 (5.9)	0 (0)
Asthma, *n* (%)	0 (0.0)	1 (5.9)
Atrial fibrillation, *n* (%)	0 (0.0)	1 (5.9)
Thyroid disease, *n* (%)	0 (0.0)	0 (0.0)
Inflammatory disease, *n* (%)	0 (0.0)	1 (5.9)
Any bowel disease, *n* (%)	2 (11.9)	0 (0.0)
Anticoagulant or antiplatelet treatment, *n* (%)	0 (0.0)	0 (0.0)
HbA1c, mmol/mol, mean (SD)	31.3 (3.1)	31.4 (2.4)

Abbreviations: *n*: number; SD: standard deviation; BMI: body mass index; HbA1c: glycated haemoglobin.

**Table 2 biomedicines-11-02052-t002:** Results from the linear mixed models. Baseline is the intercept of the regression, ΔPlacebo is the effect of placebo (time), ΔPrednisolone is the effect of prednisolone. Abbreviations: CI: confidence interval; TEG: thromboelastography; MA: maximal amplitude of clot; R: reaction time; LY30: lysis in 30 min; K: clotting time; VWF:RCo: von Willebrand factor–ristocetin cofactor activity; kIU/L: kilo international units per liter; VWF:Ag: von Willebrand factor antigen; FEU: fibrin equivalent units.

Measurement	Baseline (95% CI)	Δ Placebo (95% CI)	Δ Prednisolone (95% CI)	Missing Placebo/Prednisolone, *n* (%)
TEG:MA (mm)	61.72 (60.64, 62.80, *p <* 0.0001)	0.73 (−0.47, 1.92, *p =* 0.22)	−0.77 (−2.48, 0.94, *p =* 0.37)	0 (0.00)/2 (11.76)
TEG:R (min)	5.36 (4.82, 5.89, *p <* 0.0001)	0.14 (−0.51, 0.79, *p =* 0.66)	−0.39 (−1.25, 0.48, *p =* 0.37)	0 (0.00)/2 (11.76)
TEG:Angle (degrees)	64.43 (62.06, 66.82, *p <* 0.0001)	2.46 (0.38, 4.53, *p =* 0.02)	−1.53 (−4.10, 1.04, *p =* 0.23)	0 (0.00)/2 (11.76)
TEG:K (min)	1.78 (1.61, 1.95, *p <* 0.0001)	−0.08 (−0.25, 0.09, *p =* 0.33)	0.02 (−0.22, 0.25, *p =* 0.89)	0 (0.00)/2 (11.76)
TEG:LY30 (%)	0.60 (0.29, 0.90, *p =* 0.0003)	0.11 (−0.44, 0.66, *p =* 0.68)	0.25 (−0.56, 1.06, *p =* 0.54)	0 (0.00)/2 (11.76)
Platelet count (×10^9^/L)	231.58 (213.80, 249.35, *p <* 0.0001)	8.11 (−7.64, 23.86, *p =* 0.30)	8.65 (−14.08, 31.38, *p =* 0.44)	0 (0.00)/1 (5.88)
VWF:Ag (kIU/L)	1.10 (0.97, 1.24, *p <* 0.0001)	−0.05 (−0.16, 0.06, *p =* 0.38)	0.31 (0.15, 0.46, *p =* 0.0004)	0 (0.00)/(0.00)
VWF:RCo (kIU/L)	0.80 (0.71, 0.89, *p <* 0.0001)	−0.03 (−0.09, 0.03, *p =* 0.31)	0.15 (0.07, 0.23, *p =* 0.0006)	0 (0.00)/0 (0.00)
INR	1.07 (1.02, 1.11, *p <* 0.0001)	−0.02 (−0.05, 0.02, *p =* 0.42)	0.02 (−0.27, 0.07, *p =* 0.34)	1 (5.88)/1 (5.88)
APTT (seconds)	27.99 (26.79, 29.19, *p <* 0.0001)	0.09 (−2.11, 2.28, *p =* 0.94)	−0.47 (−3.82, 2.87, *p =* 0.78)	0 (0.00)/3 (17.65)
Prothrombin (kIU/L)	0.98 (0.93, 1.02, *p <* 0.0001)	0.009 (−0.03, 0.05, *p =* 0.63)	0.05 (−0.001, 0.11, *p =* 0.05)	0 (0.00)/1 (5.88)
Fibrinogen (µmol/L)	6.94 (6.42, 7.46, *p <* 0.0001)	0.13 (−0.36, 0.62, *p =* 0.59)	−1.02 (−1.69, −0.36, *p =* 0.004)	2 (11.76)/3 (17.65)
D-dimer (mg FEU/L)	0.26 (0.21, 0.32, *p <* 0.0001)	0.06 (0.01, 0.11, *p =* 0.01)	−0.05 (−0.12, 0.01, *p =* 0.11)	1 (5.88)/1 (5.88)
Antithrombin (kIU/L)	0.94 (0.92, 0.97, *p <* 0.0001)	0.01 (−0.02, 0.04, *p =* 0.38)	0.05 (0.01, 0.09, *p =* 0.013)	1 (5.88)/2 (11.76)
Protein C (kIU/L)	0.97 (0.89, 1.05, *p <* 0.0001)	−0.02 (−0.10, 0.07, *p =* 0.67)	0.30 (0.19, 0.41, *p <* 0.0001)	0 (0.00)/2 (11.76)

## Data Availability

The dataset underlying the conclusions of our article is available following contact to a corresponding author.

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
