# Peer review of "Effect of 10-Day Treatment with 50 mg Prednisolone Once-Daily on Haemostasis in Healthy Men—A Randomised Placebo-Controlled Trial"

_biomedicines, 2023, doi:10.3390/biomedicines11072052_

Round 1

Reviewer 1 Report

In this study, authors investigated the effect of prednisolone treatment on haemostasis.

  Study is relatively well written, however, several issues should be addressed:   - in Introduction, more information from current knowledge on the topic, including the baseline results from similar experiments from the field could be added - more information on participant enrolment should be added - how they were approached, how many declined and why etc. - table 2 caption is not descriptive - change it accordingly - Discussion is rather limited - more emphasis should be put into clinical importance of these results, and future implications/hypotheses in the field    

Reviewer 2 Report

To:

Editorial Board

Biomedicines

Title: “Effect of 10 days treatment with 50 mg prednisolone once-daily on haemostasis in healthy men – a randomized placebo-controlled trial”

Dear Editor,

I read this paper and I think that:

-          The small sample size is a limitation of the study. This should be discussed in a dedicated limitation section.

-          Please provide a post-hoc sample size calculation.

-          I could not understand the reason for administering 50 mg prednisolone in healthy individuals. This should be specified.

-          Inclusion and exclusion criteria should be better described.

Reviewer 3 Report

Although i agree with study design and planned work by authors, there are some lacking considerations that cannot be ignored: first of all underling Diseases that require the use of steroids is usually a pro thrombotic condition per se and this aspect should be remarked in their text. As per other drugs the time of exposition to prothrombotic drugs may be the best prothrombotic way (eg oral contraceptives, methotrexate , antineoplastic drugs and so on), i suggest to authors to underline this aspect in discussion. Finally in daily clinical practice patients that need steroids frequently already are taking antithrombotics (antiplatelets or anticoagulants or both) so Please add this variable in your table in order to explain that your studi ed patients are off any antithrombotic therapy.

experimental design is right so.

Round 2

Reviewer 1 Report

No further comments.

Author Response

Thank you for this comment.